# Applications of Higher-Order $q$-Derivative to Meromorphic $q$-Starlike Function Related to Janowski Function

Likai Liu [1], Rekha Srivastava [2]  and Jin-Lin Liu [3,*]

1   Information Technology Department, Nanjing Vocational College of Information Technology, Nanjing 210023, China
2   Department of Mathematics and Statistics, University of Victoria, Victoria, BC V8W 3R4, Canada
3   Department of Information Science, Yangzhou University, Yangzhou 225002, China
*   Correspondence: jlliu@yzu.edu.cn

**Abstract:** By making use of a higher-order $q$-derivative operator, certain families of meromorphic $q$-starlike functions and meromorphic $q$-convex functions are introduced and studied. Several sufficient conditions and coefficient inequalities for functions in these subclasses are derived. The results presented in this article extend and generalize a number of previous results.

**Keywords:** $q$-derivative; analytic functions; differential subordination; meromorphic functions; meromorphic $q$-starlike functions; meromorphic $q$-convex functions

**MSC:** 30C45; 05A30



## 1. Introduction

Recently, $q$-analysis has fascinated scholars due to various applications in many areas of physics and mathematics. The applications of $q$-analysis were first considered by Jackson [1,2]. In recent years, some scholars have written a number of papers [3–15] associated with $q$-starlike functions and the Janowski functions [16]. In particular, Srivastava [17,18] pointed out some applications and mathematical explanations of $q$-derivatives in GFT. In this paper, we consider several families of meromorphically multivalent $q$-starlike functions by making use of the Janowski function and the higher-order $q$-derivative. Certain sufficient conditions and coefficient inequalities for functions in these subclasses are derived. Moreover, several previous results are generalized.

Let $\Sigma(p)(p \in \mathbb{N})$ denote the family of $p$-valent analytic functions in $U^* = \{z : 0 < |z| < 1\}$ which have the following form:

$$f(z) = z^{-p} + \sum_{k=1}^{\infty} a_{k-p} z^{k-p}. \tag{1}$$

Furthermore, we let $\Sigma(1) = \Sigma$.
If $f \in \Sigma(p)$ satisfies the condition

$$\mathrm{Re}\left(-\frac{zf'(z)}{f(z)}\right) > 0, \tag{2}$$

then $f$ is called a meromorphic $p$-valent starlike function. We note the family by $M\Sigma^*(p)$ and write $M\Sigma^*(1) = M\Sigma^*$. The class $M\Sigma^*$ was studied by Pommerenke [19].
If $f \in \Sigma(p)$ satisfies the condition

$$\mathrm{Re}\left(-\frac{(zf'(z))'}{f'(z)}\right) > 0, \tag{3}$$

then $f$ is called a meromorphic $p$-valent convex function. We note this family by $MC(p)$ and write $MC(1) = MC$.

If a function $\psi$ is analytic in $U = U^* \cup \{0\}$ and satisfies

$$\psi(z) = 1 + \sum_{k=1}^{\infty} \psi_k z^k \tag{4}$$

and

$$\mathrm{Re}\big(\psi(z)\big) > 0,$$

then $\psi$ is said to be in the family $P$.

Let $\varphi$ be analytic in $U$ and $\varphi(0) = 1$. If $\varphi$ satisfies

$$\varphi(z) \prec \frac{Az + 1}{Bz + 1} \quad (-1 \leqq B < A \leqq 1),$$

then $\varphi$ is said to be in $P[A, B]$.

In [16], Janowski studied the family $P[A, B]$ and obtained that $\varphi$ is in $P[A, B]$ if

$$\varphi(z) = \frac{(1 - A) + (1 + A)\psi(z)}{(1 - B) + (1 + B)\psi(z)} \quad (\psi \in P; -1 \leqq B < A \leqq 1).$$

Let $f$ and $g$ be analytic in $U$. If there exists $w$ analytic in $U$ with $w(0) = 0$ and $|w(z)| < 1$, so that $f(z) = g(w(z))$, then we say that $f$ is subordinate to $g$, written by $f \prec g$. Further, if the function $g$ is analytic and univalent in $U$, then

$$f \prec g \quad (z \in U) \iff f(U) \subset g(U) \text{ and } f(0) = g(0).$$

**Definition 1.** *Let $f \in \Sigma$. Then $f$ is called to be in $M\Sigma^*[A, B]$ if*

$$-\frac{zf'(z)}{f(z)} = \frac{(1 - A) + (1 + A)\psi(z)}{(1 - B) + (1 + B)\psi(z)} \quad (-1 \leqq B < A \leqq 1; \psi \in P). \tag{5}$$

In [20], Karunakaran studied the family $M\Sigma^*[A, B]$.

Let $0 < q < 1$ and define $[\tau]_q$ as the following:

$$[\tau]_q = \begin{cases} \dfrac{q^\tau - 1}{q - 1} & (\tau \in \mathbb{C}) \\ \sum_{k=0}^{n-1} q^k = 1 + q + q^2 + \cdots + q^{n-1} & (\tau = n \in \mathbb{N}). \end{cases}$$

Let $0 < q < 1$. The $q$-factorial $[m]_q!$ is defined by

$$[m]_q! = \begin{cases} 1 & (m = 0) \\ \Pi_{k=1}^{m}[k]_q & (m \in \mathbb{N}). \end{cases}$$

Let $\gamma \in \mathbb{N}$. We define $q$-Pochhammer symbol $[\gamma]_{q,n}$ by (see [21])

$$[\gamma]_{q,n} = \begin{cases} 1 & (n = 0) \\ \Pi_{k=\gamma}^{\gamma+n-1}[k]_q & (n \in \mathbb{N}). \end{cases}$$

Specifically, we write $[0]_{q,n} = 0$.

Next, we define the $q$-derivative $D_q$ $(0 < q < 1)$ for $f \in \Sigma(p)$ by

$$(D_q f)(z) = \frac{f(zq) - f(z)}{z(q - 1)}$$

$$= -\sum_{k=1}^{p} \left( \frac{[k]_{q,1}}{q^k} \right) a_{-k} z^{-1-k} + \sum_{k=1}^{\infty} [k]_q a_k z^{-1+k}, \tag{6}$$

where $a_{-p} = 1$.

From (6), we can see that

$$\lim_{q \to 1} (D_q f)(z) = \lim_{q \to 1} \frac{f(zq) - f(z)}{z(q-1)} = f'(z).$$

Further, one can find that

$$(D_q^{(2)} f)(z) = \sum_{k=1}^{p} \left( \frac{[k]_{q,2}}{q^{2k+1}} \right) a_{-k} z^{-k-2} + \sum_{k=2}^{\infty} [k-1]_{q,2} a_k z^{k-2}, \tag{7}$$

$$(D_q^{(3)} f)(z) = - \sum_{k=1}^{p} \left( \frac{[k]_{q,3}}{q^{3k+3}} \right) a_{-k} z^{-k-3} + \sum_{k=3}^{\infty} [k-2]_{q,3} a_k z^{k-3}, \tag{8}$$

$$\cdots \cdots \cdots$$

$$(D_q^{(p)} f)(z) = (-1)^p \sum_{k=1}^{p} \left( \frac{[k]_{q,p}}{q^{p[k+\frac{1}{2}(p-1)]}} \right) a_{-k} z^{-k-p} + \sum_{k=p}^{\infty} [k-p+1]_{q,p} a_k z^{k-p}, \tag{9}$$

where $a_{-p} = 1$ and $D_q^{(p)}$ is called $p$th order $q$-derivatives.

**Definition 2.** *Let $f \in \Sigma$. If $f$ satisfies*

$$\left| \frac{qz D_q f(z)}{f(z)} + \frac{1}{1-q} \right| < \frac{1}{1-q}, \tag{10}$$

*then $f$ is called to be in the meromorphic $q$-starlike function family $M\Sigma_q^*$.*

It is easily seen that, when $q \to 1^-$, the disk given by (10) becomes

$$\mathrm{Re} \left( - \frac{z f'(z)}{f(z)} \right) > 0.$$

Thus, the class $M\Sigma_q^*$ reduces to the meromorphic starlike function family $M\Sigma^*$ (see [19]). Furthermore, we can rewrite (10) as the following:

$$- \frac{qz D_q f(z)}{f(z)} \prec \widehat{h}(z) \quad \text{where} \quad \widehat{h}(z) = \frac{z+1}{1-qz}.$$

Further, the meromorphic $q$-convex function family $MC_q$ could be derived by

$$f(z) \in MC_q \Leftrightarrow -qz D_q f(z) \in M\Sigma_q^*.$$

**Definition 3.** *If a function $f \in \Sigma(p)$ satisfies*

$$- \frac{q^{2p-1} z (D_q^{(p)} f)(z)}{[2p-1]_q (D_q^{(p-1)} f)(z)} \prec \frac{(A+1)\widehat{h}(z) + (1-A)}{(B+1)\widehat{h}(z) + (1-B)} \quad \left( \widehat{h}(z) = \frac{z+1}{1-qz} - 1; \; \leqq B < A \leqq 1 \right),$$

*or, equivalently,*

$$- \frac{q^{2p-1} z (D_q^{(p)} f)(z)}{[2p-1]_q (D_q^{(p-1)} f)(z)} \prec s(z), \tag{11}$$

*where*

$$s(z) = \frac{z(1+A) - zq(1-A) + 2}{z(1+B) - zq(1-B) + 2} \quad (0 < q < 1; \; -1 \leqq B < A \leqq 1), \tag{12}$$

*then $f$ is in the family $M\Sigma_q^*[p, A, B]$.*

**Remark 1.** *We write the following special cases:*

(i)    $M\Sigma_q^*[1, A, B] = M\Sigma_q^*[A, B]$, *when* $p = 1$.

(ii)   $\lim_{q \to 1^-} M\Sigma_q^*[1, A, B] = M\Sigma^*[A, B]$, *when* $p = 1$.

(iii)  $M\Sigma_q^*[p, A, B] = M\Sigma_q^*[\alpha]$, *when* $p = 1$, $A = 1 - 2\alpha$ $(0 \leqq \alpha < 1)$ *and* $B = -1$. *In [19],*
       *Pommerenke considered the family* $M\Sigma_q^*[\alpha]$ .

Now we define the meromorphic $q$-convex function family $MC_q[p, A, B]$ by

$$f \in MC_q[p, A, B] \iff \frac{(-1)^p q^{\frac{1}{2}p(3p-1)}}{[p]_{q,p}} z^p D_q^{(p)} f \in M\Sigma_q^*[p, A, B].$$

In particular, we write $MC_q[p, A, B] = MC_q[A, B]$ when $p = 1$.

**Lemma 1** ([22]). *Let* $\psi(z) = 1 + \psi_1 z + \psi_2 z^2 + \cdots$ *belong to the family P. Then*

$$|\psi_2 - \nu\psi_1^2| \leqq \begin{cases} 4\nu - 2 & (\nu > 1) \\ 2 & (0 \leqq \nu \leqq 1) \\ -4\nu + 2 & (\nu < 0). \end{cases} \tag{13}$$

**Lemma 2** ([23]). *Let* $h(z) = 1 + \sum_{k=1}^{\infty} h_k z^k$ *be analytic in U. Furthermore, let* $H(z) = 1 + \sum_{k=1}^{\infty} C_k z^k$ *be univalent convex in U. If* $h(z) \prec H(z)$, *then*

$$|h_k| \leqq |C_1| \quad (k \geqq 1).$$

## 2. Main Results

**Theorem 1.** *If*

$$g(z) = z^{-p} + \sum_{k=1}^{\infty} a_{-p+k} z^{-p+k} \in M\Sigma_q^*[p, A, B] \quad (p \geqq 2),$$

*then*

$$|a_{2-p} - \mu a_{1-p}^2| \leqq \begin{cases} \left(\frac{A-B}{2}\right) \left(\frac{[2p-3]_{q,3}}{q^{2p-2}[p-2]_{q,2}}\right) \Lambda(q) & (\mu > \sigma_1) \\ \left(\frac{A-B}{2}\right) \left(\frac{[2p-3]_{q,3}}{q^{2p-2}[p-2]_{q,2}}\right) & (\sigma_2 \leqq \mu \leqq \sigma_1) \\ \left(\frac{B-A}{2}\right) \left(\frac{[2p-3]_{q,3}}{q^{2p-2}[p-2]_{q,2}}\right) \Lambda(q) & (\mu < \sigma_2), \end{cases}$$

*where*

$$\Lambda(q) = \frac{\left\{ \begin{array}{l} \{(1 + Bq[2p-2]_q - A[2p-1]_q)(q+1) - 2\}[p-1]_q[2p-3]_q \\ + \mu(q+1)^2(A-B)[2p-2]_{q,2}[p-2]_q \end{array} \right\}}{2[p-1]_q[2p-3]_q},$$

$$\sigma_1 = \frac{[p-1]_q[2p-3]_q\{4 + (q+1)(A[2p-1]_q - qB[2p-2]_q - 1)\}}{(q+1)^2(A-B)[p-2]_q[2p-2]_{q,2}}$$

*and*

$$\sigma_2 = \frac{[p-1]_q[2p-3]_q\{(q+1)(A[2p-1]_q - 1 - qB[2p-2]_q)\}}{4(q+1)^2(A-B)[p-2]_q[2p-2]_{q,2}}.$$

**Proof.** From the assumption of the theorem, we obtain

$$-\frac{q^{2p-1} z(D_q^{(p)} g)(z)}{[2p-1]_q (D_q^{(p-1)} g)(z)} \prec \phi(z),$$

where

$$\phi(z) = \frac{2 - q(1-A)z + (1+A)z}{2 - q(1-B)z + (1+B)z}.$$

This gives that

$$-\frac{q^{2p-1}z(D_q^{(p)}g)(z)}{[2p-1]_q(D_q^{(p-1)}g)(z)} = \phi(w(z)),$$

where $w(z)$ is a Schwarz function. Now a function $h(z)$ is defined as follows:

$$h(z) = \frac{1 + w(z)}{1 - w(z)} = 1 + \sum_{n=1}^{\infty} h_n z^n \in P.$$

$\square$

Furthermore, one can see that

$$\phi(w(z)) = \frac{2(h(z) + 1 - q(h(z) - 1)) + (1+A)(q+1)(h(z) - 1)}{2(h(z) + 1 - q(h(z) - 1)) + (1+B)(q+1)(h(z) - 1)}$$

$$= 1 + \frac{1}{4}(1+q)(A-B)h_1 z + \frac{1}{16}(1+q)(A-B)\left\{4h_2 - (B(1+q) - q + 3)h_1^2\right\}z^2 + \cdots.$$

Similarly, we find that

$$-\frac{q^{2p-1}z(D_q^{(p)}g)(z)}{[2p-1]_q(D_q^{(p-1)}g)(z)} = 1 - \frac{[p-1]_q}{[2p-2]_{q,2}}q^{p-1}a_{1-p}z$$

$$+ \frac{[p-1]_q}{[2(p-1)]_{q,2}}q^{2(p-1)}\left(\frac{[p-1]_q}{[2(p-1)]_q}a_{1-p}^2 - \frac{[p-2]_q}{[2p-3]_q}(1+q)a_{2-p}\right)z^2 + \cdots$$

for $p \geqq 2$. Therefore, for $p \geqq 2$, we obtain

$$a_{1-p} = -\frac{(q+1)(A-B)[2(p-1)]_{q,2}}{4q^{p-1}[p-1]_q}h_1 \tag{14}$$

and

$$a_{2-p} = \frac{(A-B)[2p-3]_{q,3}}{q^{2p-2}[p-2]_{q,2}}\left(\frac{1}{16}k_1(q)h_1^2 - \frac{1}{4}h_2\right), \tag{15}$$

where

$$k_1(q) = (1+q)\{A[2p-1]_q - B([2p-1]_q - 1) - 1\} + 4. \tag{16}$$

Hence we obtain for $p \geqq 2$ that

$$|a_{2-p} - \mu a_{1-p}^2| = \left(\frac{A-B}{4}\right)\left(\frac{[2p-3]_{q,3}}{q^{2p-2}[p-2]_{q,2}}\right)|h_2 - k_2 h_1^2|, \tag{17}$$

where

$$k_2 = \frac{[p-1]_q[2p-3]_q k_1(q) - \mu[2(p-1)]_{q,2}[p-2]_q(A-B)(q+1)^2}{4[2p-3]_q[p-1]_q}$$

with $k_1(q)$ given by (16).

Now we can see that the conditions $\mu > \sigma_1$, $\sigma_2 \leqq \mu \leqq \sigma_1$ and $\mu < \sigma_2$ in Theorem 1 imply that $k_2 < 0$, $0 \leqq k_2 \leqq 1$ and $k_2 > 1$, respectively. By applying Lemma 1 in (17), the desired result is obtained. This proves Theorem 1.

Applying the same method as in the proof of Theorem 1, we obtain the following theorem for the case $p = 1$.

**Theorem 2.** *If*

$$g(z) = z^{-1} + \sum_{k=1}^{\infty} a_{k-1} z^{k-1} \in M\Sigma_q^*[A, B],$$

*then*

$$|a_1 - \mu a_0^2| \leqq \begin{cases} \left(\frac{A-B}{4}\right)[(1-A)q + \mu(q+1)^2(A-B) - A - 1] & \left(\mu > \frac{(A-1)q+A+3}{(A-B)(q+1)^2}\right) \\ \frac{A-B}{2} & \left(\frac{A-1}{(A-B)(q+1)} \leqq \mu \leqq \frac{(A-1)q+A+3}{(A-B)(q+1)^2}\right) \\ \left(\frac{B-A}{4}\right)[(1-A)q + \mu(q+1)^2(A-B) - A - 1] & \left(\mu < \frac{A-1}{(A-B)(q+1)}\right). \end{cases}$$

Letting $q \to 1^-$, $A = 1$ and $B = -1$ in Theorem 2, we obtain a result of the known family $M\Sigma^*$.

**Corollary 1.** *If*

$$g(z) = z^{-1} + \sum_{k=1}^{\infty} a_{k-1} z^{k-1} \in M\Sigma^*,$$

*then*

$$\left|a_1 - \mu a_0^2\right| \lessapprox \begin{cases} 4\mu - 1 & \left(\mu > \frac{1}{2}\right) \\ 1 & \left(0 \leqq \mu \leqq \frac{1}{2}\right) \\ 1 - 4\mu & (\mu < 0). \end{cases}$$

**Theorem 3.** *Let $p \geqq 2$. If*

$$g(z) = z^{-p} + \sum_{k=1}^{\infty} a_{-p+k} z^{-p+k} \in M\Sigma_q^*[p, A, B],$$

*then*

$$\left|a_{-p+k}\right| \leqq \prod_{j=1}^{k} \frac{2[j-1]_q + [2p-1]_q(q+1)(A-B)}{2[j]_q q^{p-1}[p-j]_q} \tag{18}$$

*for $1 \leqq k \leqq p - 1$.*

**Proof.** If $g$ belongs to $M\Sigma_q^*[p, A, B]$, then

$$\psi(z) := -\frac{q^{2p-1} z(D_q^{(p)} g)(z)}{[2p-1]_q (D_q^{(p-1)} g)(z)} \prec \phi(z), \tag{19}$$

where

$$\phi(z) = \frac{z(1+A) - zq(1-A) + 2}{z(1+B) - zq(1-B) + 2} = 1 - \frac{1}{2}(1+q)(B-A)z + \frac{1}{4}(1+q)(B-A)\{1 + B(1+q) - q\}z^2 + \cdots .$$

□

Let

$$\psi(z) = 1 + \sum_{k=1}^{\infty} \psi_k z^k.$$

Applying Lemma 2, we obtain

$$|\psi_k| \leqq \frac{1}{2}(q+1)(A-B) \quad (k \geqq 1). \tag{20}$$

Furthermore, from (19), we have

$$-q^{2p-1} z(D_q^{(p)} g)(z) = \{[2p-1]_q (D_q^{(p-1)} g)(z)\} \psi(z),$$

which implies that

$$- q^{2p-1} \left( \sum_{k=1}^{p} \frac{(-1)^p [k]_{q,p}}{q^{p[k+\frac{1}{2}(p-1)]}} a_{-k} z^{-p-k+1} + \sum_{k=p}^{\infty} [-p+k+1]_{q,p} a_k z^{-p+k+1} \right)$$

$$= [2p-1]_q \left( 1 + \sum_{k=1}^{\infty} \psi_k z^k \right) \left( \sum_{k=1}^{p} \frac{(-1)^{p-1} [k]_{q,p-1}}{q^{[k+\frac{1}{2}(p-2)](p-1)}} a_{-k} z^{-p-k+1} \right.$$

$$\left. + \sum_{k=p-1}^{\infty} [2+k-p]_{q,p-1} a_k z^{-p+k+1} \right),$$

where $a_{-p} = 1$.

It is easily seen from the above formula that

$$|a_{-p+k}| \leqq \frac{(A-B)(q+1)[2p-1]_q q^{\frac{1}{2}(p-1)(3p-2k-2)}}{2[k]_q [p-k]_{q,p-1}} \sum_{l=1}^{k} \frac{[p+l-k]_{q,p-1}}{q^{\frac{1}{2}(3p-2k+2l-2)(p-1)}} \left| a_{k-p-l} \right| \quad (21)$$

for $1 \leqq k \leqq p-1$.

Now

$$|a_{1-p}| \leqq \frac{[2p-2]_{q,2}(1+q)(A-B)}{2q^{p-1}[p-1]_q},$$

$$|a_{2-p}| \leqq \frac{q^{\frac{1}{2}(p-1)(3p-6)}[2p-1]_q(1+q)(A-B)}{2[p-2]_{q,p-1}[2]_q} \left\{ \frac{[p]_{q,p-1}}{q^{\frac{1}{2}(3p-2)(p-1)}} + \frac{[p-1]_{q,p-1}}{q^{\frac{1}{2}(3p-4)(p-1)}} |a_{1-p}| \right\}$$

$$= \frac{(1+q)(A-B)[2p-2]_{q,2}}{2q^{p-1}[p-1]_q} \cdot \frac{\{2+[2p-1]_q(1+q)(A-B)\}[2p-3]_q}{2q^{p-1}[2]_q[p-2]_q},$$

$$\cdots\cdots\cdots$$

$$|a_{k-p}| \leqq \prod_{j=1}^{k} \frac{2[j-1]_q + (1+q)(A-B)[2p-1]_q}{2q^{p-1}[j]_q[p-j]_q}$$

for $1 \leqq k \leqq p-1$. This proves Theorem 3.

Applying the same methods as in the proof of Theorem 3, we obtain the following Theorems 4 and 5.

**Theorem 4.** *Let* $p \geqq 2$. *If* $g(z) = z^{-p} + \sum_{k=1}^{\infty} a_{-p+k} z^{-p+k}$ *belongs to* $MC_q[p, A, B]$, *then*

$$|a_{-p+k}| \leqq \frac{[p]_{q,p}}{q^{pk}[p-k]_{q,p}} \prod_{n=1}^{k} \frac{[2p-1]_q(1+q)(A-B) + 2[n-1]_q}{2[n]_q q^{p-1}[p-n]_q}$$

*for* $1 \leqq k \leqq p-1$.

**Theorem 5.** *Let* $g(z) = z^{-1} + \sum_{k=1}^{\infty} a_{k-1} z^{k-1}$ *belong to* $M\Sigma_q^*[A, B]$. *Then*

$$|a_{k-1}| \leqq \prod_{n=1}^{k} \frac{(1+q)(A-B) + 2[n-1]_q}{2[n]_q}$$

*for* $k \geqq 2$.

Letting $q \to 1-$, $A = 1 - 2\alpha$ $(0 \leqq \alpha < 1)$ and $B = -1$ in Theorem 5, we have a result of the known family $M\Sigma^*(\alpha)$.

**Corollary 2.** *Let* $g(z) = z^{-1} + \sum_{k=1}^{\infty} a_{k-1} z^{k-1} \in M\Sigma^*(\alpha)$. *Then*

$$|a_{k-1}| \leqq \prod_{j=1}^{k} \frac{j - 2\alpha + 1}{j}$$

*for $k \geqq 2$.*

The following equivalence could help us to study the family $M\Sigma_q^*[p, A, B]$:

$$g \in M\Sigma_q^*[p, A, B] \iff \left| \frac{(1-B)\left\{ \frac{q^{2p-1}z(D_q^{(p)}g)(z)}{[2p-1]_q(D_q^{(p-1)}g)(z)} \right\} + (1-A)}{(1+B)\left( -\frac{q^{2p-1}z(D_q^{(p)}g)(z)}{[2p-1]_q(D_q^{(p-1)}g)(z)} \right) - (1+A)} - \frac{1}{1-q} \right| < \frac{1}{1-q}.$$

**Theorem 6.** *If a function $g(z) = z^{-p} + \sum_{k=1}^{\infty} a_{-p+k}z^{-p+k} \in \Sigma(p)$ satisfies*

$$\sum_{k=1}^{p} \left( \frac{[k]_{q,p-1}}{q^{\left[k+\frac{1}{2}(p-2)\right](p-1)}} \right) \left\{ \left| [2p-1]_q(1+A) - [p+k-1]_q q^{p-k}(1+B) \right| + 2[p-k]_q \right\} |a_{-k}|$$

$$+ \sum_{k=p}^{\infty} [k-p]_{q,p-1} \left\{ \left| [2p-1]_q(1+A) - [p+k-1]_q q^{2p-1}(1+B) \right| + 2[k+p]_q \right\} |a_k|$$

$$+ [2p-1]_q [p-1]_q!(2+A)|a_{p-1}| < \frac{(A-B)[p]_{q,p}}{q^{\frac{1}{2}(3p-2)(p-1)}}, \tag{22}$$

*then $g \in M\Sigma_q^*[p, A, B]$.*

**Proof.** By a simple calculation, we obtain

$$\left| \frac{(1-B)\left\{ \frac{q^{2p-1}z(D_q^{(p)}g)(z)}{[2p-1]_q(D_q^{(p-1)}g)(z)} \right\} + (1-A)}{(1+B)\left\{ -\frac{q^{2p-1}z(D_q^{(p)}g)(z)}{[2p-1]_q(D_q^{(p-1)}g)(z)} \right\} - (1+A)} - \frac{1}{1-q} \right|$$

$$\leqq \left| \frac{\{-q^{2p-1}z(D_q^{(p)}g)(z)\}(1-B) - [2p-1]_q(1-A)(D_q^{(p-1)}g)(z)}{\{-q^{2p-1}z(D_q^{(p)}g)(z)\}(1+B) - [2p-1]_q(1+A)(D_q^{(p-1)}g)(z)} + 1 \right| + \frac{q}{1-q}$$

$$= 2 \left| \frac{\left\{ \begin{array}{l} \sum_{k=1}^{p} \frac{(-1)^{p-1}[k]_{q,p-1}[2p-1]_q}{q^{(p-1)\left[k+\frac{1}{2}(p-2)\right]}} a_{-k}z^{p-k} + \sum_{k=p-1}^{\infty}[k+2-p]_{q,p}[2p-1]_q a_k z^{k+p} \\ + \sum_{k=1}^{p} \frac{(-1)^p[k]_{q,p}q^{2p-1}}{q^{p\left[k+\frac{1}{2}(p-1)\right]}} a_{-k}z^{p-k} + \sum_{k=p}^{\infty} q^{2p-1}[k+1-p]_{q,p}a_k z^{n+p} \end{array} \right\}}{\left\{ \begin{array}{l} (A-B)\frac{(-1)^{p-1}[p]_{q,p}}{q^{\frac{1}{2}(p-1)(3p-2)}} + (A+1)[p-1]_q![2p-1]_q a_{p-1}z^{2p-1} \\ + \sum_{k=1}^{p-1} \frac{(-1)^{p-1}[k]_{q,p-1}}{q^{(p-1)\left[k+\frac{1}{2}(p-2)\right]}} \{[2p-1]_q(A+1) - (1+B)q^{p-k}[p-1+k]_q\}a_{-k}z^{p-k} \\ + \sum_{k=p}^{\infty} \{[2p-1]_q(A+1) - (B+1)q^{2p-1}[p-1+k]_q\}[k-p]_{q,p+1}a_k z^{k+p} \end{array} \right\}} \right|$$

$$+ \frac{q}{1-q}$$

$$\leqq 2 \frac{\sum_{k=1}^{p-1} \frac{[k]_{q,p-1}[p-k]_q}{q^{(p-1)\left[k+\frac{1}{2}(p-2)\right]}}|a_{-k}| + [p-1]_q![2p-1]_q|a_{p-1}| + \sum_{k=p}^{\infty}[k-p]_{q,p+1}[k+p]_q|a_k|}{\left\{ \begin{array}{l} \left( \frac{[p]_{q,p}}{q^{\frac{1}{2}(p-1)(3p-2)}}(A-B) - (1+A)[p-1]_q![2p-1]_q|a_{p-1}| \right) \\ - \sum_{k=1}^{p-1} \frac{[k]_{q,p-1}}{q^{\left[k+\frac{1}{2}(p-2)\right](p-1)}} \left| \{[2p-1]_q(A+1) - (B+1)[p+k-1+]_q q^{p-k}\} \right| |a_{-k}| \\ - \sum_{k=p}^{\infty}[k-p]_{q,p+1} \left| \{[2p-1]_q(A+1) - (B+1)q^{2p-1}[p+k-1]_q\} \right| |a_k| \end{array} \right\}} + \frac{q}{1-q}. \tag{23}$$

$\square$

Now, it follows from (22) that the last expression in (23) is less than $\frac{1}{1-q}$. This proves Theorem 6.

**Theorem 7.** *If a function* $g(z) = z^{-1} + \sum_{k=1}^{\infty} a_{k-1} z^{k-1} \in \Sigma$ *satisfies*

$$\sum_{k=1}^{\infty} \left\{ 2[k]_q + \left| q[k-1]_q(1+B) - (1-A) \right| \right\} |a_{k-1}| < A - B, \tag{24}$$

*then* $g \in M\Sigma_q^*[A, B]$.

**Proof.** By simple calculation, we have

$$\left| \frac{(1-B)\left(\frac{qz(D_q g)(z)}{g(z)}\right) + (1-A)}{(1+B)\left(-\frac{qz(D_q g)(z)}{g(z)}\right) - (1+A)} - \frac{1}{1-q} \right|$$

$$\leqq \frac{q}{1-q} + 2\left| \frac{g(z) + qz(D_q g)(z)}{q(1+B)z(D_q g)(z) + (1+A)} \right|$$

$$= \frac{q}{1-q} + 2\left| \frac{\sum_{k=1}^{\infty} [k]_q |a_{k-1}|}{(A-B) - \sum_{k=1}^{\infty} [q(1+B)[k-1]_q + (1+A)]|a_{k-1}|} \right|. \tag{25}$$

From (24) we can see that (25) is less than $\frac{1}{1-q}$. This proves Theorem 7. □

Letting $A = 1 - 2\alpha$ ($0 \leqq \alpha < 1$), $B = -1$ and $q \to 1^-$ in Theorem 7, we obtain a result of the known family $M\Sigma^*(\alpha)$.

**Corollary 3.** *If a function* $g(z) = z^{-1} + \sum_{k=1}^{\infty} a_{k-1} z^{k-1} \in \Sigma$ *satisfies*

$$\sum_{k=1}^{\infty} (1 + k - \alpha)|a_{k-1}| < 1 - \alpha \quad (0 \leqq \alpha < 1),$$

*then* $g \in M\Sigma^*(\alpha)$.

Applying the same method as in the proof of Theorem 7, we obtain the following theorem.

**Theorem 8.** *If a function* $g(z) = z^{-1} + \sum_{k=1}^{\infty} a_{k-1} z^{k-1} \in \Sigma$ *satisfies*

$$\sum_{k=1}^{\infty} \left\{ 2[k]_q + \left| q[k-1]_q(1+B) - (1-A) \right| \right\} [k-1]_q |a_{k-1}| < A - B,$$

*then* $g \in MC_q[A, B]$.

**Author Contributions:** Every authors contribution is equal. All authors have read and agreed to the published version of the manuscript.

**Funding:** The work was supported by Natural Science Foundation of Nanjing Vocational College of Information Technology (Grant No.YK20180403).

**Data Availability Statement:** Not applicable.

**Acknowledgments:** The authors thank the referees for their careful reading and suggestions.

**Conflicts of Interest:** The authors declare no conflict of interest.

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
