# Peer review of "Applications of Higher-Order q-Derivative to Meromorphic q-Starlike Function Related to Janowski Function"

_axioms, doi:10.3390/axioms11100509_

Round 1

Reviewer 1 Report

I do not recommend the publication of the Manuscript ID: axioms-1886010 titled “Applications of Higher-Order q-Derivatives to Meromorphic q-Starlike Functions Related to the Janowski Functions” authored by Likai LIU, Rekha Srivastava, and JINLIN LIU.

The article presents no new knowledge and the contents can, at best, be categorized as a senior undergraduate or junior graduate work.

My rationale for this negative recommendation is attached.

Author Response

Dear the Reviewer,

We have checked and revised our paper according to the reviewers' comments. We add the Acknowledgements in the text.  Please see the attachment. 

Best  regards

Jin-Lin Liu

Reviewer 2 Report

See attached file.

Author Response

Dear the reviewer,

Many thanks for your care of our paper. We have checked and revised the paper according to your kind suggestions.

Best regards

Jin-Lin Liu

Reviewer 3 Report

Please see the attachment for details.

Author Response

Dear the Reviewer,

Many thanks for your care of our paper. We have checked and revised the paper according to your kind suggections. 

Best regards

Jin-Lin Liu

Reviewer 4 Report

 I strongly recommend that this paper be published in your journal.

Author Response

Dear the Reviewer,

Many thanks for your care of our paper. We have checked and revised the paper according your kind suggestions.

Best regards

Jin-Lin Liu

Round 2

Reviewer 1 Report

This article is very low in substance, but the results are correct. If the editorial board feel that the journal that this article is going to be published in is at the level of this article, it can be published.

The coefficient bounds presented in the theorems are unrealistic, very narrowly defined, and of very little interest to the mathematical community.